

# Pollinator identity and behavior affect pollination in kiwifruit (*Actinidia chinensis* Planch.)

Melissa A. Broussard[1], Brad G. Howlett[2], Lisa J. Evans[3], Heather McBrydie[1], Brian T. Cutting[3], Samantha F.J. Read[2] and David E. Pattemore[1,4]

[1] The New Zealand Institute for Plant and Food Research Ltd, Hamilton, New Zealand
[2] The New Zealand Institute for Plant and Food Research Ltd, Lincoln, New Zealand
[3] Plant and Food Research Australia Ltd, Brisbane, Australia
[4] University of Auckland, Auckland, New Zealand

## ABSTRACT

Many crop plants rely on insect pollination, particularly insect-pollinated crops which are functionally dioecious. These crops require insects to move pollen between separate plants which are functionally male or female. While honey bees are typically considered the most important crop pollinator species, many other insects are known to visit crops but the pollination contribution of the full diversity of these flower visitors is poorly understood. In this study, we examine the role of diverse insect pollinators for two kiwifruit cultivars as model systems for dioecious crops: *Actinidia chinensis* var. *deliciosa* 'Hayward' (a green-fleshed variety) and A. *chinensis* var. *chinensis* 'Zesy002' (a gold-fleshed variety). In our round-the-clock insect surveys, we identified that psychodid flies and mosquitoes were the second and third most frequent floral visitors after honey bees (*Apis mellifera* L), but further work is required to investigate their pollination efficiency. Measures of single-visit pollen deposition identified that several insects, including the bees *Leioproctus* spp. and *Bombus* spp. and the flies *Helophilus hochstetteri* and *Eristalis tenax*, deposited a similar amount of pollen on flowers as honey bees (*Apis mellifera*). Due to their long foraging period and high pollen deposition, we recommend the development of strategies to boost populations of *Bombus* spp., *Eristalis tenax* and other hover flies, and unmanaged bees for use as synergistic pollinators alongside honey bees.

## INTRODUCTION

Pollination by insects is an important service to global agriculture, with one third of crops relying to some extent on insect-vectored pollination (*Aizen et al., 2009*), and increasing dependence on pollinating insects worldwide (*Garibaldi et al., 2011*). Pollination is particularly critical for insect-pollinated crops which are functionally dioecious, as they require pollen to be moved from a functionally-male plant to a functionally-female plant.

*Garibaldi et al. (2013)* showed, for a range of crop species, that fruit set increased with wild pollinator visitation regardless of honey bee abundance. Later work has shown that

Corresponding author
Melissa A. Broussard,
melissa.broussard@plantandfood.co.nz,
melissa.a.broussard@gmail.com

pollinator richness is particularly important for overcoming pollination deficits in fields larger than 2 ha (*Garibaldi et al., 2016*). This effect is critical as the crops examined in the reviews all rely on managed honey bees for pollination; the synergistic effect of multiple pollinator species was, in all cases, a pollination service provided by the surrounding landscape. These wild pollinators need not be the most abundant species, either; at the landscape scale, the majority of pollinating species (even rare ones) can be critical for maintaining pollination services across farms and orchards (*Winfree et al., 2018*). At both the global and local scales, dependency on honey bees for pollination carries significant risk (*Willis & Kirby, 2015*). This risk, and the risk of wild pollinator decline, has in some places been mitigated by artificial pollination, but the practice can be costly in both labour and capital (*Hii, 2004*). To address these issues, further examination of the potential of non-honey bee pollinators is warranted.

Kiwifruit is one of a handful of major crops for which insect-vectored pollination is essential (*Klein et al., 2007*), and over 4.25 million tonnes of the fruit is produced worldwide each year (*FAOSTAT, 2018*). The climbing vines bear conspicuous cream-coloured flowers, with staminate and pistillate flowers on separate plants (*Schmid, 1978*). Unlike many other dioecious species, neither sex produces nectar; female plants instead produce flowers with anthers containing pseudopollen (*Schmid, 1978*).

Because insects must cross from the male vine to the female vine to pollinate kiwifruit, insect movement can also influence pollination success, and the orchard layout and management can influence this movement of pollinators. In kiwifruit, *Jay & Jay (1984)* reported that the orchard pruning system affected insect movement, with 76% of honey bees moving within-row in a T-bar trained orchard, compared with 64% in an orchard grown on a pergola trellis system. A number of studies also find that seed set declines with distance from male vines (*Testolin, 1991*; *Goodwin, Ten Houten & Perry, 1999*), suggesting that pollen carryover (the amount of pollen transported from flower to flower) may be an important metric in determining optimal planting distances.

The primary focus of kiwifruit pollination literature has been on honey bees, with considerably less known about other insects. Problems with the supply of honey bees in Italy has led to decades of artificial pollination, though this is generally less effective than insect pollination (*Sáiz et al., 2019*). Honey bees dominate kiwifruit pollination in the United States (*McKay, 1978*), France (*Vaissière et al., 1990*), Australia (*Howpage, Spooner-Hart & Vithanage, 2001*), and New Zealand (*Clinch, 1984*; *Howlett et al., 2017b*), but other locations (*Sharma, Mattu & Thakur, 2013*; *Miñarro & Twizell, 2015*), including China (*Steven, 1988b*; *Steven, 1988a*; *Yang & Wu, 1990*), have more diverse pollinator fauna. These other pollinator taxa can be more efficient visitors, and provide a substantial portion of total pollination services.

This complementarity effect is possibly due to insects visiting at different times-of-day, during different temperatures, or under different weather conditions. Several studies have found that the types of insect visitors vary with time-of-day and weather conditions (*Macfarlane & Ferguson, 1983*; *Howlett et al., 2013*; *Miñarro & Twizell, 2015*; *Howlett et al., 2017b*), while *Miñarro & Twizell (2015)* found that insects also vary in their flower handling time and single-visit pollen deposition, with some being more efficient pollinators than

honey bees. As well as providing significant potential for complementary pollination services, pollinator diversity can reduce an over-reliance on a single species. This is particularly pertinent in the case of gold-fleshed kiwifruit cultivars which appear to have lower pollen requirements (*Goodwin, McBrydie & Taylor, 2013*; *Goodwin et al., 2017*; *Broussard et al., 2021*): less efficient insects could have a higher proportional impact where fewer pollen grains are required as each visit would deliver a higher proportion of the total required pollen, while highly efficient insects may deliver more pollen than necessary.

It is possible to assess the effectiveness of different pollinators through single-visit pollen deposition, flower-handling time, and abundance (*King, Ballantyne & Willmer, 2013*), particularly as kiwifruit has been well-studied in terms of pollination requirements. The pollination requirements of green-fleshed kiwifruit like 'Hayward' can be met in a single honey bee visit (*Donovan & Read, 1992*), but may require up to 40 visits (*Goodwin & Haine, 1995*), while yellow-fleshed kiwifruit requires only six visits (*Goodwin et al., 2017*). Not all visits are alike; insect behavior has been shown to affect pollinator efficiency and effectiveness. For example, insects might not touch the stigmas while visiting ('illegitimate visitors') or may only visit either male or female flowers and thus not move viable pollen to female flowers (*Howlett, Lankin-Vega & Pattemore, 2015*) Such insects may not deposit any viable pollen even if they are frequent flower visitors. However, the efficiency and behavior of many alternate pollinators of kiwifruit is currently unknown, as is their relative importance for different cultivars with differing pollination requirements.

To address these gaps in knowledge around pollen requirements and the potential role of alternate pollinators of kiwifruit, we assessed:
1. Single-visit pollen deposition by diverse pollinating insects in a green-fleshed ('Hayward') and gold-fleshed ('Zesy002') cultivar,
2. Pollen carryover for bees visiting 'Zesy002',
3. Pollinator flower-handling behavior in 'Hayward' and 'Zesy002', and
4. How pollinator activity and diversity varies with time-of-day.

## MATERIALS & METHODS

We examined pollination in two common kiwifruit cultivars, *Actinidia chinensis* var. *deliciosa* 'Hayward' and A. chinensis var. chinensis 'Zesy002'. Three general assessments were done for each cultivar: single-visit pollen deposition, flower-handling, and activity patterns throughout the day.

### General methodology
We conducted all trials in New Zealand kiwifruit orchards between October and March (Austral spring and summer). All orchards trained their kiwifruit vines on a pergola system. In general, experiments were restricted to days with fine weather (15–30 °C, wind below 15 km/h), however conditions outside of this ideal were examined in round-the-clock surveys due to it being night.

Information on pollinator flower handling and visitation behavior was recorded while standing at a distance of at least 1 m from observed insects to avoid disturbing them.

## Single-visit pollen deposition

The amount of pollen deposited in a single bee visit (single-visit deposition; SVD) was assessed for both 'Hayward' and 'Zesy002'. The amount of pollen deposited in successive visits to female flowers (pollen carryover) was also determined for 'Zesy002'.

To prevent flower visits prior to SVD assessment, we bagged flowers at the popcorn stage prior to stigmas being exposed. Once the flowers were visited, we excised stigmas immediately into 1.5-mL Eppendorf tubes containing 0.5 mL of Alexander's stain (*Alexander, 1980*). This stain allows visual separation of male (green-blue) and female (purple-red) pollen (*Goodwin & Perry, 1992*). The number of pollen grains on the stigmas was then estimated using a haemocytometer as described by *Cutting et al. (2018)*.

### Single-visit pollen deposition for 'Hayward'

In 2013 and 2014, we measured SVD for insects visiting 'Hayward' kiwifruit in a Waikato orchard. SVD measurements were taken for insect movements from male to female flowers, and female to female flowers. This was done for honey bees and as many other flower-visiting insect species as could be found in the orchard. Previously bagged female flowers were removed and presented to an insect using the 'active' approach described by *Howlett et al. (2017a)*.

### Single-visit pollen deposition and carryover for 'Zesy002'

Bumble bee SVDs were assessed in 2016–2017 in a Bay of Plenty 'Zesy002' orchard near Te Puke (37.786°S, 176.325°E), which was selected because a section of the orchard could easily be isolated into a block where honey bees were not intentionally introduced during flowering. Bumble bee colonies were placed in the orchard with pollen dispensers as described in *Cutting et al. (2018)*. Groups of virgin flowers were un-bagged and either observed until some flowers received single visits from bumble bees, or flowers were picked and presented to bees. A subset of exposed virgin flowers did not receive bumble bee visits and were collected as controls. Collected stigmas were placed in Alexander's stain and the pollen grains counted.

In 2018, a second set of bumble bee SVDs was obtained in a different Bay of Plenty 'Zesy002' orchard near Te Puke using the same method. Unlike the 2016-2017 trial, this orchard was covered with hail netting (ceiling and sides), and colonies were not fitted with pollen dispensers.

Single-visit pollen deposition and pollen carryover was assessed for honey bees on November 6–8, 2018 in a Waikato 'Zesy002' orchard near Cambridge (37.891°S, 175.465°E). Up to three previously bagged unvisited flowers were simultaneously loaded into a pollination block and presented to bees foraging on male flowers. Bees were allowed to visit flowers in the block sequentially, and the order visited was recorded along with the bee identity.

## Pollinator flower handling

Pollinators' flower-handling time and frequency of switching between male and female vines was assessed in both 'Hayward' and 'Zesy002'. Individual insects were followed and

their movement from flower to flower was followed. The sex of the visited flower, insect contact between stigmas and anthers, and time spent on flowers were recorded.

### Pollinator Flower Handling for 'Hayward'

We recorded pollinator flower visiting behavior over a 3-year period (2013–2015), encompassing 20 orchards across New Zealand, including nine in the Bay of Plenty, 8 in Gisborne, three in Hawke's Bay, and three in Nelson/Tasman. Insects were observed visiting kiwifruit flowers during daylight hours (0845 h –1540h), and behavior observations recorded. To maximize the diversity of insect species recorded, observations were made throughout the day and recordings were alternated between species that were present. Up to 15 min was spent after each recording to search for species that had not recently been recorded.

### Pollinator flower handling for 'Zesy002'

Following the same methodology, audio recordings of honey bee and bumble bee activity were made in 2016 in a Bay of Plenty 'Zesy002' orchard near Te Puke (37.786°S, 176.325°E). The number of flowers and duration of each visit was transcribed for each insect followed. In the same year at another Bay of Plenty orchard, near Aongatete (37.609°S, 175.945°E), flower handling time and bee was recorded for honey bees. The time taken for a honey bee to visit two flowers was measured with a stop watch (start: land on first flower, end: land on third flower) in the morning (between 1000 h and 1200h) and afternoon (between 1500 h and 1600h).

## Pollinator activity patterns

Pollinator activity was assessed using an instantaneous measure of insects per 1,000 flowers; this metric was chosen over bees per flower per hour insects per 1,000 flowers is the primary measure in kiwifruit pollination literature, with studies finding that a point-in-time measure of six honey bees per 1,000 flowers corresponded to a full pollination (*Palmer-Jones, Clinch & Briscoe, 1976*; *Clinch, 1984*; *Goodwin, 1987*; *Donovan & Read, 1992*; *Goodwin & Haine, 1995*; *Pomeroy & Fisher, 2002*; *Goodwin, McBrydie & Taylor, 2013*; *Goodwin et al., 2017*).

## Pollinator activity patterns for 'Hayward'

Surveys of insect visitors to kiwifruit flowers were conducted in 2013 and 2014 in one Waikato kiwifruit orchard near Morrinsville (37.657°S, 175.525°E). Each year, a total of 25 counts of insects along transects were taken between 0400 h and 2200 h between November 22 and December 3.

Each transect involved a timed 10-min walk down a single row with multiple transects conducted simultaneously. All open female flowers observed directly by the observers were counted using a handheld counter. All insects observed visiting flowers were noted, and a second hand-held counter was used when necessary for honey bees. Total insect counts per transect were divided by the number of open female flowers observed and then multiplied by 1,000 to estimate flower visitor numbers per 1,000 flowers.

*Pollinator activity patterns for 'Zesy002'*

In 2016 at a Bay of Plenty orchard near Aongatete (37.609°S, 175.945°E), bee density were recorded for honey bees. The number of flowers was counted in a quadrat, and the number of bees was counted in the same quadrat every 2 h between 0900 h and 1700 h on 7 days between 2 and 8 November. These values were then used to calculate the density of bees per 1,000 female flowers at each time point.

## Statistical analysis

Data analysis was done in R (*R Core Team, 2014*). As none of the datasets met assumptions of normality, nonparametric tests were chosen. Comparisons of means were conducted using Mann–Whitney U (MWU) tests. Comparisons of multiple groups were done with Kruskal-Wallis tests, and pairwise testing of Kruskal-Wallis test results with Bonferroni-corrected Dunn tests. For examining the nonlinear patterns of activity of particular taxa (or groups of taxa) over a 24-hour period, we utilized generalized additive mixed models (GAMMs) in the R package gamm4 (*Wood & Scheipl, 2009*) with the number of insects per 1,000 flowers as the response variable, time-of-day as the predictor, and year and sample date as nested random effects. Models were tested for over-dispersion.

## RESULTS

### Single-visit pollen deposition for 'Hayward'

In total, SVDs were collected from 622 insects belonging to 12 species. The most commonly collected non-honey bee visitors were flower longhorn beetles (*Zorion guttigerum*), March flies (*Dilophus nigrostigma*), and bumble bees (*B. terrestris* and *B. hortorum/ruderatus*). Pollen deposition varied considerably amongst the taxa observed ($P < 0.001$ Kruskal-Wallis test; Fig. 1; Table S1).

### Single-visit pollen deposition and carryover for 'Zesy002'

We assessed the SVD of 43 bumble bees and three incidental honey bees moving from female to female flowers in 2016. On average, *Bombus terrestris* from colonies fitted with pollen dispensers deposited $16,953 \pm 4,401$ staminate pollen grains (standard error of the mean [SEM]) onto a 'Zesy002' flower (Fig. 2). Eight of these bees (16%) deposited fewer than 100 pollen grains. None of the three honey bees deposited more than 100 grains.

We examined a further 16 bumble bee SVDs in a netted orchard in 2018. Six visits were from male to female flowers, with bumble bees depositing $7,241 \pm 2,164$ (SEM) staminate grains. Ten female-to-female visits were recorded, with $1,652 \pm 1,159$ (SEM) deposited.

In 2018, we recorded 96 honey bees visiting one or more female flowers presented to them in the pollination block. Pollen deposition was extremely variable, so while there was a trend toward fewer grains being deposited over successive visits, it was not statistically significant with our sample size (Fig. 3, $P = 0.805$, $z = -0.248$, GLMM). For the first visit from a male to female flower, honey bees deposited $15,961 \pm 2,459$ (SEM) pollen grains, not significantly different from deposition in 'Hayward' ($P = 0.085$, $W = 3283$, MWU).

Only 5% of first visits deposited no pollen, increasing to 7% for the second female flower. For first visits, 17.8% deposited fewer than 1,000 pollen grains.

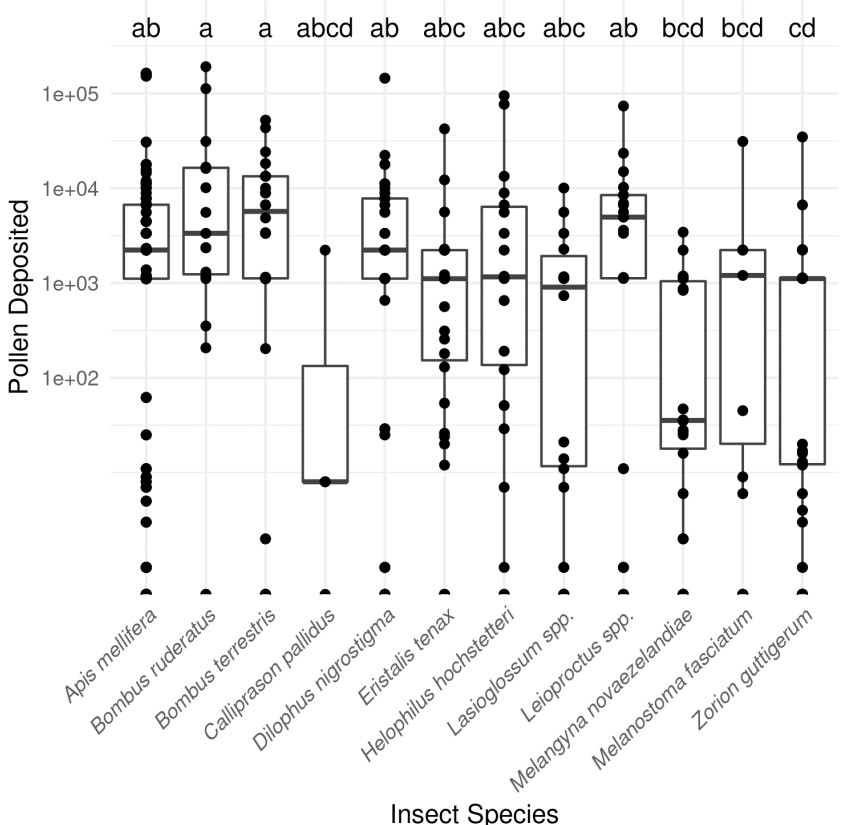

**Figure 1** **Male to female flower single-visit pollen deposition varied across insect taxa in kiwifruit cultivar** *Actinidia chinensis* var. *deliciosa* **'Hayward' between 2013 and 2015.** Boxes represent the middle 50% of the data, bars within boxes represent the median and whiskers are the spread of the data within 1.5×the interquartile range; data points outside this range are shown as dots. Letters correspond to Kruskal-Wallace groupings for species with five or more insects observed.

## Pollinator flower handling for 'Hayward'

Between 2013 and 2015, we followed 177 insects belonging to 14 species foraging on 1783 'Hayward' flowers. Insects were followed for 3.8 ± 0.3 min (SEM; range 16 s –19.8 min). The majority of insects were followed across five or more flowers, with up to 51 sequential visits observed.

*Bombus terrestris* had the shortest flower handling time, and bees in general were quicker to switch between flowers than flies or beetles ($P < 0.001$, $\chi^2 = 40.00$, $df = 2$, Kruskal-Wallis rank-sum test, Fig. 4; Table S2).

Overall, longer flower handling time for insects in this survey did not translate to higher pollen deposition for male to female (Fig. 5) or female to female visits (Table S2). However, native bees in the genus *Leioproctus*, the syrphid fly *Helophilus hochstetteri,* and the March fly *Dilophus nigrostigma* all had both high pollen deposition and long flower handling time.

Although the data suggest that there may be species-level differences in the number of switches between flower sexes per minute (Table 1), our statistical analysis did not have sufficient power to detect a difference amongst high background variance.

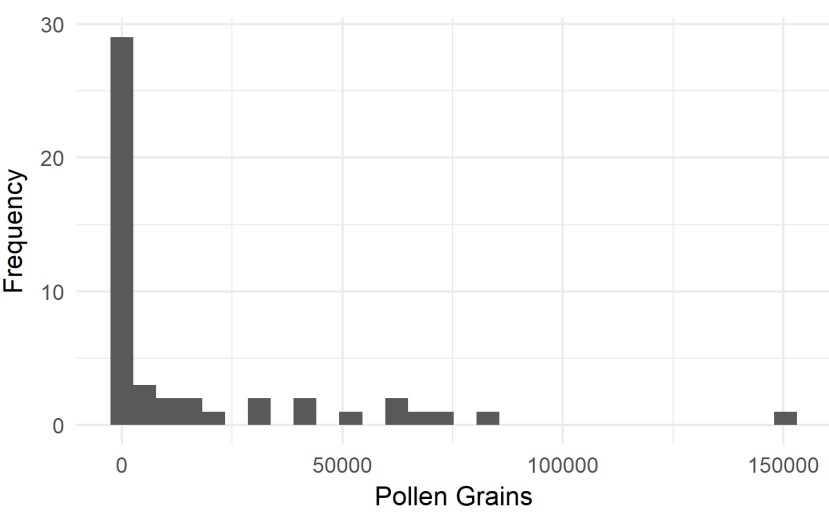

**Figure 2 Single-visit pollen deposition by *Bombus terrestris* on flowers of kiwifruit cultivar *Actinidia chinensis* var. *chinensis* 'Zesy002'.** These bees were foraging from colonies fitted with pollen dispensers.

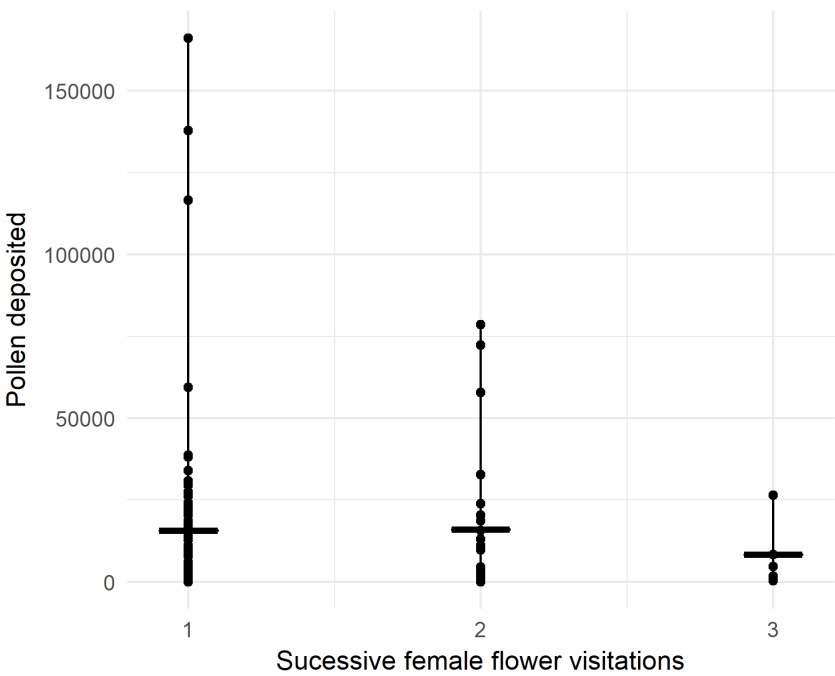

**Figure 3 Number of pollen grains honey bees deposited on sequential flower visits to previously bagged female kiwifruit cultivar *Actinidia chinensis* var. *chinensis* 'Zesy002' flowers after visiting male flowers in 2018.** Lines represent the median and whiskers represent the spread.

## Pollinator flower handling for 'Zesy002'

Bumble bees (*B. terrestris*) were observed visiting kiwifruit flowers in 134 foraging bouts in 2015, each observation averaging 8.7 ± 0.5 flowers (SEM, range 1–37). The mean

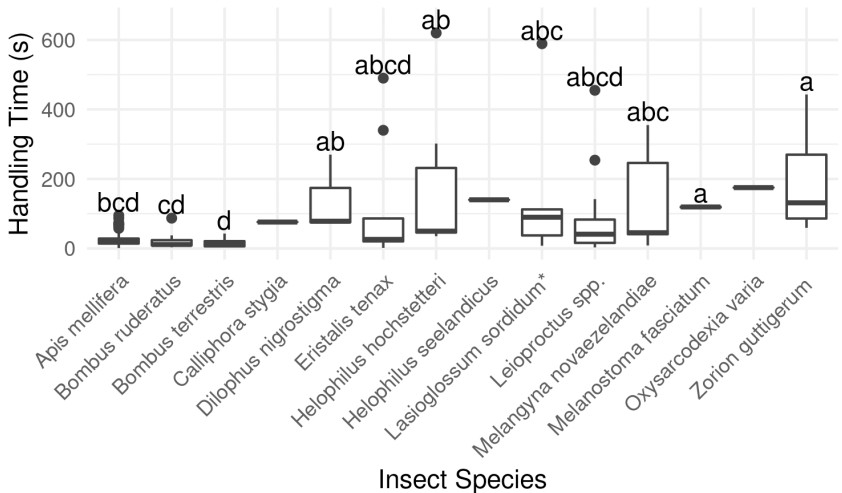

**Figure 4 Average duration of visit to single kiwifruit cultivar *Actinidia chinensis* var. *deliciosa* 'Hayward' flowers by individuals of 14 insect species between 2013 and 2015.** Boxes represent the middle 50% of the data, bars within boxes represent the median and whiskers are the spread of the data within 1.5×the interquartile range; data points outside this range are shown as dots. Letters correspond to Kruskal-Wallace groupings for species with five or more insects observed.

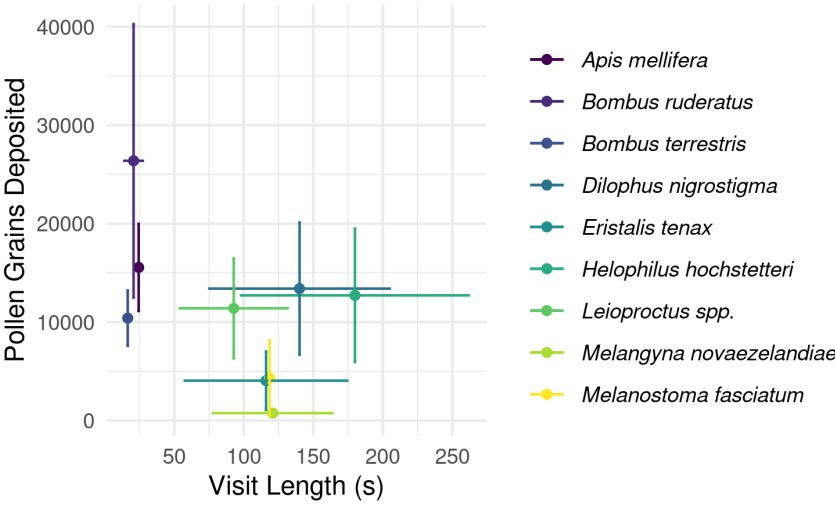

**Figure 5 Relationship between flower handling time and the number of pollen grains deposited for different pollinating insect species recorded visiting kiwifruit cultivar *Actinidia chinensis* var. *deliciosa* 'Hayward' in 2013–2015.** Points are species-level means for flower handling time and the number of pollen grains deposited on a male to female visit. Bars around points represent the standard error.

per-flower handling time did not significantly differ in bouts that did or did not include male flowers ($P = 0.390$, $W = 1676.5$, MWU); the average flower handling time for a bumble bee was $11.5 \pm 0.3$ s (weighted SEM). Weighting the results by number of flowers visited, 27.8% of bumble bees switched between female and male flowers.
**Table 1  Proportion of insects switching between flower sexes in kiwifruit cultivar *Actinidia chinensis* var. *deliciosa* 'Hayward' between 2013 and 2015 for insect species that visited at least 20 flowers, weighted by flower and time.** Proportion switched is the proportion of total individuals observed that switched between sexes at least once during the observation period (*Aizen et al., 2009*). Switches are reported as weighted mean ± standard error. Letters are based on the Dunn test for insects with >1 observation.

| Insect | n | Observation (min) | Flowers | Proportion switched | Switches/ flower | Switches/ minute | Group |
|---|---|---|---|---|---|---|---|
| **Hymenoptera** | | | | | | | |
| *Apis mellifera* | 86 | 268.46 | 957 | 0.31 | 0.17 | 0.16 ± 0.05 | a |
| *Bombus ruderatus* | 12 | 25.91 | 122 | 0.08 | 0.07 | 0.04 ± 0.19 | a |
| *Bombus terrestris* | 20 | 49.67 | 309 | 0.20 | 0.17 | 0.10 ± 0.06 | a |
| *Lasioglossum* spp. | 6 | 23.40 | 23 | 0.17 | 0.11 | 0.04 ± 0.05 | a |
| *Leioproctus* spp. | 12 | 36.49 | 78 | 0.17 | 0.81 | 0.14 ± 0.22 | a |
| **Diptera** | | | | | | | |
| *Eristalis tenax* | 9 | 49.54 | 128 | 0.44 | 0.44 | 0.24 ± 0.17 | a |
| *Helophilus hochstetteri* | 7 | 53.63 | 38 | 0.29 | 0.09 | 0.04 ± 0.06 | a |
| *Melangyna novaezelandiae* | 9 | 45.07 | 45 | 0.22 | 0.41 | 0.09 ± 0.15 | a |
| *Melanostoma fasciatum* | 4 | 37.33 | 23 | 0.50 | 0.13 | 0.13 ± 0.12 | a |
| **Coleoptera** | | | | | | | |
| *Zorion guttigerum* | 6 | 68.32 | 40 | 0.33 | 0.08 | 0.06 ± 0.03 | a |

In the 2016 videos of pollinators, 1291 honey bees were recorded foraging in open orchards. The flower handling time was 16.1 ± 0.4 s (SEM). Bees visiting 1- to 2-day old, white-colored flowers spent slightly longer foraging (16.7 ± 0.4 s SEM) than those visiting 3- to 4-day old, golden-colored flowers (13.2 ± 0.8 s SEM; $P < 0.001$, MWU).

A total of 52 honey bees were timed visiting two flowers in 2016, 28 in the morning, and 18 in the afternoon. The average time taken to visit and travel between two flowers was 32.3 ± 2.3 s (SEM). The single-visit equivalent was 16.2 ± 1.4 s (SEM), which was considerably shorter than the 50.9 ± 7.3 s (SEM) it took a honey bee to forage on a previously bagged flower. There was no difference in visit length in the morning and afternoon ($P = 0.396$, $W = 351$, MWU).

## Pollinator activity patterns for 'Hayward'

In our around-the-clock pollinator surveys in 2013 and 2014, we observed 5,867 insects visiting kiwifruit flowers, 75.8% of which were honey bees. The next most common floral visitors were much less abundant: 724 (12.3%) were moth flies (Diptera: Psychodidae), 107 (1.8%) were *Zorion guttigerum* (Coleoptera: Cerambycidae), and 84 (1.4%) were mosquitoes (Diptera: Culicidae). For non-*Apis* bees, *B. terrestris* represented 0.3% of visitors, and the native *Lasioglossum* spp. and *Leioproctus* spp. together represented less than 0.2%. Syrphid flies together represented 0.5% of floral visitors. Of the four most abundant taxa, there was a time-of-day effect for two (Fig. 6), which was corroborated by a GAMM for *Apis mellifera* ($P < 0.001$, $F = 46.2$) and Culicidae ($P < 0.001$, $F = 11.53$). Psychodidae ($P = 0.057$, $F = 2.77$) and *Zorion guttigerum* ($P = 0.051$, $F = 3.064$) exhibited trends toward nocturnal and diurnal activity, though these were not statistically significant at the Bonferroni-corrected $\alpha$ of 0.013. While our sample sizes were not large enough to do species-level analysis for the remaining taxa, we found that time-of-day was a good

predictor of the abundance of other beetles ($P = 0.006$, $F = 4.986$, GAMM) and syrphid flies ($P = 0.002$, $F = 5.895$, GAMM). Non-*Apis* bees tended to forage in the middle of the day, though their low abundance made this trend not significant at the corrected $\alpha$ ($P = 0.023$, $F = 2.774$, GAMM). Other flies exhibited a bimodal distribution, peaking at dawn and dusk, though again this was not significant at our sample size ($P = 0.018$, $F = 3.025$, GAMM, Fig. 6).

## Pollinator activity Patterns for 'Zesy002'

Honey bees were observed foraging between 0900 h and 1700 h. Bees were more abundant at 1100 h than at other times ($P < 0.001$, $W = 26506$, MWU), and least abundant at 1700 h ($P < 0.001$, $W = 14540$, MWU, Fig. 7). The median number of bees / 1,000 female flowers was 3.8, and reached 40 bees / 1,000 flowers at numerous time points. Several outlier data points, where there were few flowers in the randomly selected quadrat and high bee numbers in the bay resulted in calculations of bees / 1,000 flowers which ranged between 80 and 300—much higher than has previously been reported in the literature (*Palmer-Jones, Clinch & Briscoe, 1976*; *Clinch, 1984*; *Goodwin, 1987*; *Goodwin & Ten Houten, 1988*). We therefore only used data from bays with at least 500 flowers.

## DISCUSSION

Functionally dioecious crops are highly reliant on insect-mediated pollination to produce a marketable crop. Kiwifruit, like many agricultural crops, is primarily pollinated by honey bees, but we found that a diversity of other insects may be able to significantly contribute to kiwifruit pollination, with several species as efficient as honey bees.

As kiwifruit can produce fruit regardless of the time-of-day pollination occurs (*Broussard et al., 2021*), pollination need not be limited to peak honey bee foraging hours (late morning to early afternoon). There is the potential for other pollination strategies to compliment managed honey bees, including alternative insect pollinators and artificial pollination. Honey bees may leave the orchard in the afternoon due to low pollen supply (*Goodwin, 1987*), but the anthers of kiwifruit flowers still contain some pollen at this time of day, and floral visitors able to utilize smaller quantities of pollen may assist in transfer in the late afternoon. Nocturnal pollinators also have potential, even with higher pollen requirements as flower opening occurs at night and early in the morning (*Thakur & Rathore, 1991*; *Goodwin, McBrydie & Taylor, 2013*). *Howlett et al. (2017b)* identified that a number of taxa maintained abundance into the afternoon, including the bees *Leioproctus* spp., *Lasioglossum* spp, and *Bombus terrestris*, flower flies (*Eristalis tenax* and *Helophilus* spp.), March flies (*Dilophus nigrostigma*), and flower longhorn beetles (*Zorion guttigerum*). Likewise, *Macfarlane & Ferguson (1983)* found that bumble bees, *Leioproctus*, syrphid flies (including *Eristalis tenax*) and March flies were still visiting flowers into the evening (1700h), and that grass moths (Crambidae) visited flowers at night.

Insect visits varied substantially in the amount of pollen deposited. We found that seven taxa were not statistically different from honey bees in their SVD: *Bombus terrestris*, *B. hortorum/ruderatus*, *Leioproctus* spp., *Lasioglossum* spp., *D. nigrostigma*, *H. hochstetteri*, and *E. tenax*. Some of these taxa (*e.g.*, the small sweat bee *Lasioglossum* spp.) had low average

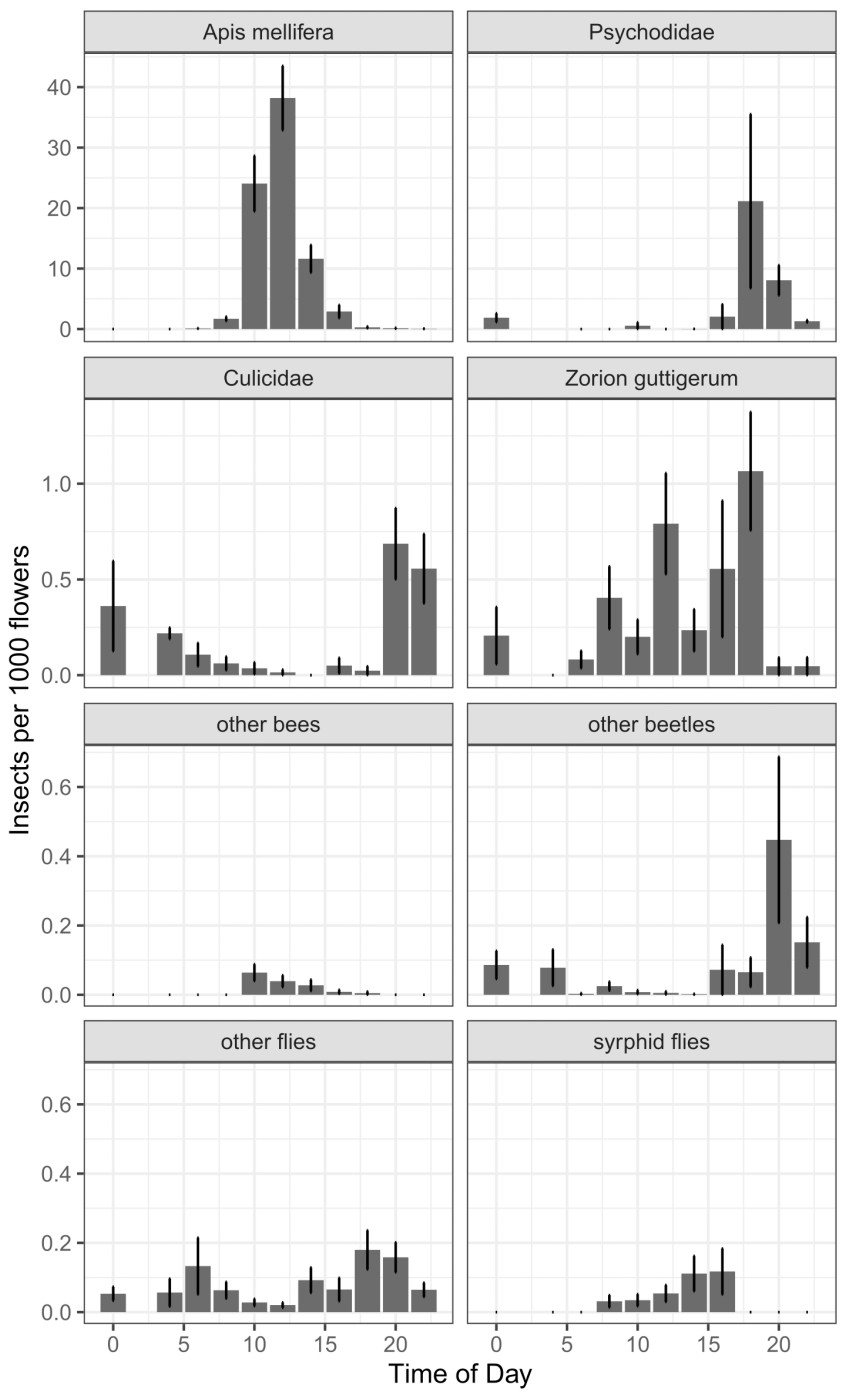

**Figure 6** Number of insects per 1000 open female flowers observed at different times of day for the most abundant taxa found foraging on kiwifruit cultivar *Actinidia chinensis* var. *deliciosa* 'Hayward' between 2013 and 2014. Time is recorded as hours after midnight. Bars represent mean, error bars represent standard error.

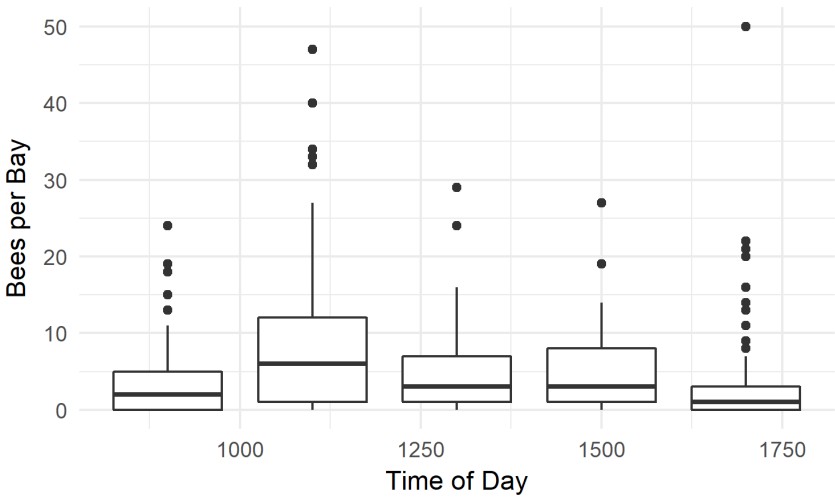

**Figure 7** **Time of day honey bees were observed foraging on kiwifruit cultivar *Actinidia chinensis* var. *chinensis* 'Zesy002'.** Values are an aggregate of 7 days' observations in 2016. Boxes represent the middle 50% of the data, bars within boxes represent the median and whiskers are the spread of the data within 1.5x the interquartile range; data points outside this range are shown as dots.

pollen deposition (but high variability), and further sampling is likely to find it has a lower SVD than honey bees. For other taxa, we have confidence in our findings as *Macfarlane & Ferguson (1983)* also reported that several *Leioproctus* species were substantially more efficient than honey bees, as was *B. terrestris*. Both honey bees and *B. terrestris* deposited about the same amount of pollen to 'Zesy002' and 'Hayward'. The *B. terrestris* colonies in 'Zesy002' orchards fitted with pollen dispensers deposited about twice as much as unassisted bumble bees deposited in open 'Hayward' orchards and more than twice as much as bumble bees in a netted "Zesy002" orchard without pollen dispensers. While some caution should be taken directly comparing these different study designs, it appears that fitting bumble bee colony boxes with pollen dispensers may improve their pollen deposition in kiwifruit orchards if this is a limiting factor.

Other authors report that a larger proportion of *B. terrestris* touch kiwifruit stigmas than honey bees (48 v 25% (*Macfarlane & Ferguson, 1983*); 86 v 67.5% (*Miñarro & Twizell, 2015*)), which may be why we found a higher proportion of bumble bees depositing pollen compared to honey bees. Bumble bees have been noted previously to be effective pollinators of kiwifruit (*Macfarlane & Ferguson, 1983*; *Pomeroy & Fisher, 2002*; *Miñarro & Twizell, 2015*; *Cutting et al., 2018*), and steps to improve their management and cost in orchards could reduce the current reliance on honey bees as a single managed pollinator. Additionally, as more orchards move toward covered cropping, it may be beneficial to incorporate bumble bees into pollination regimes, as honey bees are negatively affected by overhead netting (*Evans et al., 2019*).

*Miñarro & Twizell (2015)* note that *Eristalis* spp. and predatory hoverflies touch stigmas less frequently than bees (35% and 33.8% stigma contact). In our dataset, *E. tenax* and the predatory hover flies *Melangyna novaezelandiae* and *Melanostoma fasciatum* form a cluster

of high flower-handling time and low average pollen deposition. However, *E. tenax* does not deposit significantly less than honey bees with our current sample sizes, and along with *Lasioglossum* spp., *H. hochstetteri,* and *D. nigrostigma* is equivalent to honey bees. Given that all three fly species forage into the evening, they may be promising candidates for supplemental pollination that have not yet been explored, particularly as methods are being developed to mass-rear *E. tenax* for pollination (*Howlett & Gee, 2019*), which may be possible to apply to *H. hochstetteri*, which also has a rat-tailed maggot larval stage. It is also potentially worth examining creating landscapes which promote *Dilophus nigrostigma* populations, as this fly species' reasonably high (but variable) pollen deposition was also reported by *Macfarlane & Ferguson (1983)*, which increases the likelihood that the high pollen deposition seen in this study is a reliable observation.

Of note is that gold-fleshed kiwifruit appear to have a lower pollination requirement than green kiwifruit (*Broussard et al., 2021*), with green-fleshed 'Hayward' requiring 8x the insect visits of yellow-fleshed varieties 'Hort16A' and 'Zesy002' (*Broussard et al., 2021*). This means that non-*Apis* pollinators, even those who deposit less pollen per visit, may have a greater part to play in these cultivars, as the number of grains they do deposit represents a larger percentage of the overall requirement. Examining the pollination efficiency of these other taxa in yellow-fleshed kiwifruit will help assess whether any of the candidates we have highlighted here are promising candidates for supplemental pollination in those cultivars.

There are a number of insect species which have the potential to complement honey bees in kiwifruit pollination, as they are active at different times of day and are effective pollinators. More work should be done examining the frequent nocturnal flower visitors (mosquitoes and moth flies) to determine if they are depositing pollen. Encouraging other insects through landscape features or by introducing managed non-honey bee pollinators may be a good way to ensure an abundance of varied pollinators as an insurance against pollinator losses and potentially obtain synergistic increases in crop pollination.

## CONCLUSIONS

While a number of alternate pollinators of kiwifruit are known from countries around the world (*Steven, 1988a*; *Steven, 1988b*; (*Yang & Wu, 1990*; *Macfarlane & Ferguson, 1983*; *Sharma, Mattu & Thakur, 2013*; *Miñarro & Twizell, 2015*), pollination of this dioecious crop is nearly totally reliant on honey bees, a significant risk in the longer term (*Willis & Kirby, 2015*). We found a variety of insects visiting kiwifruit flowers, varying in their behavior and pollination efficiency. Unmanaged insects, both bees (*Leioproctus* spp. and *Bombus* spp.) and flies (*Helophilus hochstetteri* and *Eristalis tenax*) were as efficient as honey bees, and are promising alternative pollinators, and if their populations can be increased in orchards, they would increase redundancy and stability in pollination services. We also report a large number of nocturnal flower visitors, including psychodid flies and mosquitoes –these were the second and third most frequent floral visitors after honey bees. The pollination efficiency of these species has not been examined, but the abundance of these insects means that it is quite possible that they are performing significant pollination services. Additionally, the activity patterns of a number of flower

visiting taxa are complimentary to the patterns of honey bees –an orchard which has many of these pollinating species could see synergistic increases in pollination; our data recommend the use of *Bombus* spp., *Eristalis tenax* and other hover flies, and unmanaged bees in particular.

## ACKNOWLEDGEMENTS

Milena Janke, Tamatea Nathan, Theo Van Noort, Max Buxton, Grant Fale, Sarah Cross, Elizabeth Bull, Crystal Felman, Michelle Taylor, Alexandre Benoist, Helene Le Chenadec, Rachael L'helgoualc'h, Simon Cornut, Murielle Cuenin, Philomene Brunelliere, Thomas Besnier, Miguel Peterle, and all the landowners who graciously allowed us to conduct research on their properties. Thanks also to Paul Martinsen, Ruth Williams, and Warrick Nelson and two anonymous reviewers who provided feedback on the manuscript.

### Funding

David E. Pattemore was awarded the Plant & Food Research Discovery Science grant DS 14-65 for examining time-of-day effects. David E. Pattemore and Bradley G. Howlett received funding for work on bumble bees and wild pollinators through the New Zealand Ministry for Business, Employment & Innovation ("MBIE", www.mbie.govt.nz/) under grant no. C11X1309, "Bee minus to Bee plus and Beyond: Higher Yields from Smarter, Growth-focused Pollination Systems". Zespri Group Ltd provided funding for projects led by LE (GP1700, as part of Ministry for Primary Industries (https://www.mpi.govt.nz/) Sustainable Farming Fund project no. 404958; and GP1723, efficiency of bumble bee pollination of 'Zesy002' as part of MBIE C11X1309) and David E. Pattemore (GP1976, pollination of 'Zesy002' by bumble bees as part of MBIE C11X1309). Zespri Group Ltd contracted PFR to conduct the research projects independently to address questions of importance to their business, and the other granting agencies had no role in the research process apart from approving the initial proposal and granting funding. The funders had no role in study design, data collection and analysis, decision to publish, or preparation of the manuscript.

### Grant Disclosures

The following grant information was disclosed by the authors:
The Plant & Food Research Discovery Science: DS 14-65.
The New Zealand Ministry for Business, Employment & Innovation: C11X1309.
The New Zealand Ministry for Primary Industries: 404958.

### Competing Interests

The authors declare there are no competing interests. Melissa A. Broussard, Brad G. Howlett, Samantha F Read and David E. Pattemore are employed by The New Zealand Institute for Plant & and Food Research Limited and Lisa J. Evans and Brian T. Cutting are employed by Plant and Food Research Australia Ltd.

## Author Contributions

- Melissa A. Broussard conceived and designed the experiments, performed the experiments, analyzed the data, prepared figures and/or tables, authored or reviewed drafts of the paper, and approved the final draft.
- Brad G. Howlett, Lisa J. Evans, Brian T. Cutting and David E. Pattemore conceived and designed the experiments, performed the experiments, authored or reviewed drafts of the paper, and approved the final draft.
- Heather McBrydie and Samantha F.J. Read performed the experiments, authored or reviewed drafts of the paper, and approved the final draft.

## Data Availability

The raw data are available in the Supplemental File.

## Supplemental Information

Supplemental information for this article can be found online at http://dx.doi.org/10.7717/peerj.12963#supplemental-information.

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
