# Peer review of "Pollinator identity and behavior affect pollination in kiwifruit (Actinidia chinensis Planch.)"

_PeerJ, doi:10.7717/peerj.12963_

## Round 0.1 · original submission · Major Revisions

We have now received 2 reviews. A reject and minor revision. Based on my own reading of the ms, I advise major revisions to be returned to the reviewers for another round of review. This paper does contribute to our understanding of pollination in kiwifruit. However, it cannot be published in its current form. To add to the reviewers' comments:

Include significance values on the graphs.

There are too many figures, remove Fig. 1 and 6. Fig. 9 can just be reported in the text. Fig. 7 is too similar to table 2, either keep the figure or the table.

L193-195 and relevant graphs: why 1000 flowers? Rather use the standard metric of visits per flower per hour.

It is important to clearly report the total observation hours, as well as the number of flowers observed for the various parts of the study.

Methods: have a separate analyses section. Since some stats methods are reported, but some are not reported or not clear. Then clearly explain the different statistical analyses conducted.

Reviewer 1 ·

Basic reporting

I commend the authors for the effort of putting together a collection of experiments on kiwifruit pollinators conducted in New Zealand in the last years. However, the different designs, methodology, sites and cultivars in which the experiments were conducted, and the small sample sizes in some cases make it difficult to obtain robust conclusions from this research.

One interesting point is the visit of non-honeybee pollinators in the 24h study. And I will put it as one of the most important of these studies.

Another one interesting point is the quantity of grey literature used in the discussion, and I encourage the authors to write a review paper on pollination of kiwifruit by insects using all this bibliography.

Experimental design

The exposition of methodology is not clear nor easy to follow, and some effort should be done for clarity. Methods should be re-structured according to the exposition of the objectives.

In some experiments you say the site name, sometimes not.

L149-150. ‘fully netted’. I do not understand well.

L164… Is it necessary to detail that some observations were recorded in a device instead of written on paper?

Validity of the findings

The conclusions are not very robust for the intrinsic limitations commented above.

Additional comments

Title: I cannot see clear the effect on pollination as the pollination service (fruit-set/seed-set, fruit size) was assessed in just one experiment (with honeybees).

I will re-structure introduction from general to particular points, starting with a) importance of pollination, b) importance of pollinator diversity, c) kiwifruit as dependent of insect pollination, d) what is known about diversity of pollinators and pollination service in kiwifruit (knowledge gaps), e) hypothesis/objectives. A weakness of the introduction is that there is not a hypothesis or a clearly-stated objective, and the necessity of this research is not strongly justified.

L94-98. You have not explained nothing about cultivars yet, so they should not appear as part of the objectives.

Some tables and figures are repetitive (showing the same results), e.g. Tab1& Fig2, Tab2&Fig7. One of them could be deleted.

Fig 6 could be also deleted as it does not show relevant information.

Fig 11 and 12 could be pooled in just one

Tables 1-3. Not all readers will know those insects by the scientific names. It would be useful to add Order and maybe Family.

L280. I could not find statistical results for handling time.

L283. Why despite? I would expect: more handling time, more pollen deposition

Discussion. Can you really talk about efficiency of pollinators without measures of pollination service? Could pollen deposition be seen as a measure of such service? More justification is needed

L385-387 (and Tab1). Maybe there are no significant differences (given to the Bonferroni adjustment??), but do you really think that Lasioglossum (mean: 1955) deposit the same number of pollen grains than B. hortorum (mean: 26381)?

Reviewer 2 ·

Basic reporting

The English language used throughout the paper is clear, and unambiguous. The introduction and background showed the context well. Obviously, the literature cited in this paper were arranged well and are relevant to the central topic. The paper is structured well, and figures are relevant.

Experimental design

Research questions were well defined and listed. The investigation performed by the authors, were rigorous. The methods are standard. Methods in this paper was described with sufficient detail and information, except the ‘data analysis’ section was missing.

Validity of the findings

The conclusions in this paper are well stated, linked to original research question and limited to supporting results.

Additional comments

General comments
In this paper entitled ‘Pollinator identity and behavior affect pollination in kiwifruit (Actinidia chinensis Planch.)’, the authors investigated pollinator efficacy of multiple insect faunas in a score of kiwifruit orchards. The background of this study was interesting and meaningful because kiwifruit is an insect-pollinated crop which is functionally dioecious, so finding out the real efficient pollinator would contribute to the production of this crop. Another significance of this study, is that this paper provided a direct and rigid evidence that kiwifruit is pollinated by insects rather than wind, as anecdotally reported previously. The major finding of this paper is clear and simple, that the authors found many insects other than honeybees performed similar efficacy (in the paper showed as single visit pollen deposition, SVD). This result is meaningful, though not surprising. If the authors could evaluate single visit pollen removal (SVR) in the future, they may find that bees would remove so much pollen from the flower but deposit so little pollen on stigmas, thus bees might not be as efficient as other observed insect pollinators if consider the ratio of deposition/removal. There is also a point that the authors may consider in future studies: the authors recommended to boost populations of Bombus species due to the SVD results showed in this paper, however, despite the ability of depositing similar number of pollen grains with honeybees, bumblebees are also very efficient in collecting pollen grains from flowers, thus reduce possible pollen dispersal of plants, resulting in a waste of male gametes of plants.
Generally, the experiment has cost a lot of labor. The results and conclusions of this paper is clear and convincible. A big flaw is that the authors did not provide any information about their data analysis. Anyway, it’s proper for this paper to be published on this journal.
While there are a few minor comments which may help to improve the quality of the manuscript if adopted:
L106 does this sentence mean there exist some ‘impossible’ situation? Is pollination studied in bad weather conditions in some cases?
L198 How were data analyzed? It’s not mentioned in the materials and methods section.
L344 The detail of data analysis is missing, so the occurrence of ‘GAMM’ is abrupt.

---

## Round 0.2 · accepted · Accept

I am writing to inform you that your manuscript - Pollinator identity and behavior affect pollination in kiwifruit (Actinidia chinensis Planch.) - has been Accepted for publication